# Co-Correcting: Combat Noisy Labels in Space Debris Detection

**Hui Li** **, Zhaodong Niu \*, Quan Sun and Yabo Li**

National Key Laboratory of Science and Technology on ATR, College of Electronic Science and Technology, National University of Defense Technology, Changsha 410073, China

\* Correspondence: niuzd@nudt.edu.cn

**Abstract:** Space debris detection is vital to space missions and space situation awareness. Convolutional neural networks are introduced to detect space debris due to their excellent performance. However, noisy labels, caused by false alarms, exist in space debris detection, and cause ambiguous targets for the training of networks, leading to networks overfitting the noisy labels and losing the ability to detect space debris. To remedy this challenge, we introduce label-noise learning to space debris detection and propose a novel label-noise learning paradigm, termed Co-correcting, to overcome the effects of noisy labels. Co-correcting comprises two identical networks, and the predictions of these networks serve as auxiliary supervised information to mutually correct the noisy labels of their peer networks. In this manner, the effect of noisy labels can be mitigated by the mutual rectification of the two networks. Empirical experiments show that Co-correcting outperforms other state-of-the-art methods of label-noise learning, such as Co-teaching and JoCoR, in space debris detection. Even with a high label noise rate, the network trained via Co-correcting can detect space debris with high detection probability.

**Keywords:** space debris detection; label-noise learning; weakly supervised learning

---

## 1. Introduction

### 1.1. Background

Space debris is defined as man-made artifacts which are non-functional, comprising pieces and sections thereof. The ultimate source of space debris is the launch of different items from Earth. Launchers have grown to be more powerful and, in many circumstances, send more than one satellite into orbit. Each launch generally puts several tons of material into orbit. Most space debris is of tiny size, including payload shrouds, adapter rings, explosive bolts, instrument covers, etc., which are liberated by satellites [1]. Slag particles formed by solid rocket engines, microscopic paint flakes off surfaces, and thermal insulation blankets also play a big role in space trash. As for larger-sized debris, most of them are formed by explosions, including rocket upper stages, auxiliary engines, and satellites. Kinetic anti-satellite (ASAT) weapon tests also generate significant amounts of space debris, and even form a dangerous debris field, increasing the risk of collision with satellites [2,3]. By 2019, more than 139 million pieces of space debris had been detected [4].

A handful of enormous space debris would fall from the sky, but it is worth noting that space debris presents tremendous hazards to space missions. Due to the substantial comparative velocity of space debris, even with little size, they can release huge amounts of power after contact with launches. Thus, it is vital to locate space debris to estimate its motions and prevent collisions with it. Space debris detection is proposed to achieve this goal.

In Low Earth Orbit (LEO), most space debris is s bigger than 20 cm and can be observed by radars and optical telescopes. In 2014, the Space Surveillance and Tracking (SST) Support Framework was established by the European Union to mitigate the risk of space debris, and European ground-based radar systems were applied for space debris monitoring [5–7]. However, in Geosynchronous Earth Orbit (GEO), optical telescopes are

favored for detecting space debris. The debris in GEO has less reflection area owing to its considerable distance from sensors. In the captured optical images, the space debris merely covers a few pixels. For example, in a $256 \times 256$ size image, the space debris contains less than 0.15% pixels [4]. With low reflectance and sparse distribution, space debris appears small and dim. Instead, stars are luminous and occupy most of the area of images. Flicker noise and damaged pixels widely exist in the background. As a consequence, the signal-to-noise ratio (SNR) is incredibly low, even equivalent to 1, leading to difficulties in detecting debris from background. In short time exposure of telescopes, space debris has similar optical properties to stars, and is cataloged as stars by mistake. The ambiguous catalogs cause the noisy labels in space debris detection and pose huge challenge in training effective networks.

### 1.2. Related Work

Recently, space debris has attracted a huge amount of research interest, and debris detection is therefore proposed as preprocessing for the later operation of locating potential danger and avoiding collision with space debris.

#### 1.2.1. Classical Methods

In original observation, space debris is small and dim. Numerous traditional approaches have been developed to perform small and dim target detection. In [8], a technique based on the maximum likelihood ratio is employed to identify space debris.

Three-dimensional matched filter [9,10] and dynamic programming algorithms [11,12] are also widely utilized in small and dim object detection to cope with low SNR radar or optical images. These methods achieve great performance in small and dim object detection. However, in space debris detection, debris and stars have similar optical features, and these methods cannot separate debris from stars. As a result, more approaches were developed to remove stars from original observation.

Space debris is much closer than stars to the ground-based optical telescopes. If the mode of telescopes is set to "staring stars mode", in the field of telescopes, the stars remains stable, and the space debris moves with expected motion. In this observation mode, stars appear point-like, but debris has two different representations, acquired with the different exposure time. If the exposure time is long enough compared with the velocity of the debris, the debris appears streak-like; instead, debris becomes point-like. If the mode of telescopes is set to "staring target mode", the situation alters. The telescopes track the space debris with its expected motion. In the field of telescopes, space debris remains stable and stars keep moving instead. The debris appears point-like, but stars become point-like or streak-like based on the exposure time.

The different optical patterns of debris and stars make it possible to remove stars in observation. For example, numerous methods were adopted in Streak-like debris detection [13–15] and point-like debris detection [16,17]. The setting of telescopes and exposure time can lead to different representation of stars and debris. These methods need this prior knowledge and are only applicable to specific observation tasks. In short time exposure, debris and stars share similar features; such methods cannot achieve satisfactory performance.

Star catalogs contain the location of cataloged stars. By matching star catalogs and observation, the location of stars in observation is acquired, and then these located stars can be removed [18].

This method faces some challenges. In original observation, we can get the location of stars by matching star catalogs to images. However, with the location of stars, they cannot be removed completely without the corresponding shape, leading to residual of removal. These residuals become the main source of false alarms in debris detection. The second challenge is that the number of stars in star catalogs does not match it in observation images. For example, Gaia DR2 is a star catalog published by DPAC (Data Processing and Analysis Consortium) on 25 April 2018 with around 1.7 billion objects. Some stars in observation are not cataloged. On the other hand, some cataloged stars may be missed

by telescopes, due to their weak brightness and flickering. What is most important is that the procedure of matching stars catalogs and observation is complex. These methods require prior information of ground-based telescopes, and the procedure of stars matching is complex.

To get superior performance in background star removal, the classical subtraction approach has been frequently employed owing to its short time consumption. The purpose of the subtraction approach is to reduce the interference of stars [19]. In [20–23], a mask is applied to remove all background stars in the search area. In [24], a detection pipeline is presented to increase detection ability for faint objects by utilizing filtering and mathematical morphology. In [25], an optical masking approach named EAOM (effect analysis of the optical masking) is presented to identify space debris in the GEO region. These presented approaches adopt inter-frame difference to remove stars and perform well; however, in an actual engineering context, most stars have the property of flickering and undulating across successive frames. As a consequence, these stars cannot be totally eliminated by frame subtraction, and the leftovers comprise the primary component of false alarms.

These classical methods utilize extra prior information to remove stars before space debris detection. In good light condition and with enough exposure, debris and stars have different optical features, and these above classical methods can achieve great performance. However, in short exposure, debris and stars become similar, and stars are hard to remove from the background.

### 1.2.2. Machine Learning Methods

To avoid the procedure of star removal and construct a one-stage detection pipeline, most methods adopt a CNN (Convolutional Neural Network) into space debris detection. Due to the excellent representation abilities of CNN towards small and dim targets, these methods achieve great performance in space debris detection. In [26], a technique utilizes long short term memory (LSTM) networks to obtain high performance in recognizing and tracking small and dim objects. In [27], a YOLO-based (You Only Look Once [28]) approach is suggested, and the results demonstrate that such method is superior to classical methods such as Hough transform. In [29], Faster R-CNN with the backbone of ResNet-50 is implemented to create a detection pipeline. These methods gain substantial performance in detecting small and dim space debris. Such a performance relies heavily on massive training samples with highly accurate labeling. In good light condition and enough exposure, debris and stars have different optical features, and such methods perform well. Instead, these methods achieve unsatisfactory results when debris and stars share similar features. Once the dataset contain significant noisy labels, the networks tend to overfit the misleading direction and output incorrect predictions.

### 1.2.3. Label-Noise Learning

The outstanding performance of machine learning based methods, e.g., CNN, largely relies upon the huge size of the dataset and high accuracy of annotation. Nevertheless, annotating large-scale datasets with high precision is costly and time-consuming [30]. When the light condition is weak and exposure is insufficient, the SNR is low, and space debris and stars have similar optical properties, the extracted space debris from observation contains large amount of noisy labels. It requires experts to examine the whole dataset and select the samples with inaccurate labels (i.e., space debris with the label of stars), which would cost significant time. Another viable technique is to implement preproceedings to totally eliminate the influence of stars. However, the pipeline will grow complex, and additional prior knowledge about space debris and optical telescopes is required. The star removal methods are introduced to reduce the interference of stars, but the stars cannot be removed completely.

To minimize the cost of data cleaning, noisy samples serve as a compromise, and label-noise learning is adopted to utilize the noisy samples to train deep neural networks. Noisy samples are defined as data with ambiguous labels. For machine learning, the labels

are supervised information, which is crucial to the networks' training. Once the labels include ambiguous annotations, the supervised information becomes untrustworthy. Label-noise learning (LNL) aims to utilize noisy samples as the training data and avoid networks overfitting noisy samples. Memorization effects can explain the networks' unsatisfactory performance in noisy labels. Latest discoveries reveal that DNN overfits noisy samples during the late stage of training. However, in the early stage, DNN can recognize clean samples by itself [31]. In other words, the networks have the capacity to distinguish the samples with genuine labels at the early stage. However, the networks are degraded by the noisy labels and lose their corresponding ability.

Many researchers utilize noisy samples as training data to alleviate networks overfitting towards noisy labels. Some approaches employ regularization terms to prevent DNN overfitting towards noisy labels [32,33]. However, regularization bias [34] occurs in both explicit and implicit regularization. The estimating transition matrix is also a hotspot in label-noise learning. In this method, the label transition matrix is estimated by adding a non-linear layer built on top of softmax [35] to simulate the transition process between clean labels and noisy labels. Unfortunately, it is rather difficult to estimate such a transition matrix. To reduce complexity, most techniques assume that the transition matrix is class independent and instance independent [32,36,37]. The instance-dependent matrix is commonly assumed in most approaches as well [38].

Some representation techniques concentrate on picking reliable samples, e.g., Mentornet [39] trains an auxiliary network with the small-loss policy and then the auxiliary network picks samples for the main network. To prevent error accumulation in DNN, Co-teaching proposes to employ two similar networks to choose samples for each other [40]. In the early training stage, two networks in Co-teaching can preserve their variety owing to random initialization of networks' parameters. However, they will converge to a consensus with epoch increasing and lose the diverse learning capacity that is fundamental to "Co-teaching" paradigm. The "Update by disagreement" strategy suggested by Decoupling [41] can successfully retain the diversity of two identical networks during training. Based on this, Co-teaching+ combines Co-teaching with the "update by disagreement" strategy to slow down two networks achieving a consensus [34]. Co-teaching+ first chooses the examples with different predictions, and then these chosen instances are filtered by the small-loss policy. As a consequence, only a tiny fraction of examples is employed for training, and the performance of networks ultimately degenerates owing to inadequate training samples. For this situation, JoCoR doubts the requirement of the "Disagreement" policy to label-noise learning. Instead, JoCoR tries to maximize the agreements between two networks by combining training with Co-regularization [30].

The preceding strategies can successfully avoid networks overfitting towards noisy labels, but none of them are implemented in space debris detection. In this paper, we propose a new label-noise learning paradigm called "Co-correcting", and apply "Co-correcting" to space debris detection, avoiding manual data-cleaning and sophisticated preprocessing. Co-correcting can correct the noisy labels with auxiliary supervised information, where noisy labels provide insufficient supervised information, and trained networks can successfully detect space debris from the background.

### 1.3. Solution and Contributions of This Paper

If telescopes are used in short time exposure, debris and stars share similar optical features. In space debris detection, the similarity makes it difficult to detect space debris from the background. Classical methods utilize star catalogs, prior information of stars and debris, or inter-frame difference to remove stars in observation images. These star removal procedures are performed before detecting space debris. The stars cannot be removed completely, and the procedures of stars removal are complex. To construct a one-stage pipeline, CNN was adopted to detect debris in observation images without removing stars. The input image of CNN is the original observation, containing lots of stars. In fact, the star removal procedures implicitly exist in preparing datasets. In training stage

of CNN, the networks try to learn the inherent properties of debris and stars, and the labels are forwarded to networks as the supervision. In other words, the networks need to know the true category of training samples. The space debris and stars in datasets are assigned corresponding labels; therefore debris is separated from stars. This procedure can be accomplished by human labor or other star removal methods. Due to the inherent similarity between debris and stars, the datasets contain huge amount of noisy labels, and these noisy labels are hard to clean. Based on these points, we introduce label-noise learning into space debris detection. Label-noise learning argues that the noisy labels also reflect the inherent characteristic of debris detection, and utilizes the noisy labels to train networks according to a specific optimization policy. The star detection and removal procedures implicitly exist in the optimization of networks.

Given the aforementioned challenges, this study presents a novel label-noise learning paradigm called "Co-correcting", and applies "Co-correcting" to space debris detection. Concretely, Co-correcting adopts two identical networks and mutually corrects the original targets of noisy labels with the predictions of peer networks. In this way, Co-correcting can explicitly share supervised information across two networks and correct the noisy labels using the auxiliary supervised information. In space debris detection, we extract objects from images as space debris, and random patches from images as background. Note that the extracted space debris contains noisy labels (e.g., stars labeled as space debris). Then, the extracted sub-figures serve as training data and are sent to Co-correcting to train networks. In the inference step, the space debris can be detected by the trained networks.

The main contributions of this article are summarized as follows:

1.  We proposed a novel label-noise learning paradigm, termed Co-correcting, to train networks by directly using the data with noisy labels. Empirical results exhibit the excellent performance of Co-correcting compared to other state-of-the-art methods in label-noise learning.
2.  We are the first to introduce label-noise learning into space debris detection, and take noisy samples as a compromise to train networks. In our pipeline, the noisy training samples are directly sent into Co-correcting, therefore time-consuming manual data cleaning is avoided.

### 1.4. Organization of This Article

This paper is organized as follows. Section 2 firstly introduces the mathematical formulation of space debris and label-noise learning, and then presents the pipeline of our work, as well as the algorithm steps of our proposed method. In Section 3, we present the experiment settings and the results with detailed interpretations. In section 4, we discuss the results and analyze the performance of the methods. Lastly, Section 5 finishes this study and covers the further applications of the method.

## 2. Materials and Methods

The original observation comprises space debris, stars, hot pixels, and flicker noise, which are the forms of interference in space debris detection. In datasets, the training samples contain noisy labels, caused by ambiguous objects in observation, including stars, spots caused by cosmic rays, etc. We propose a novel label-noise learning paradigm, Co-correcting, to utilize the samples with noisy labels to train networks, and the space debris can be detected by the trained networks.

The whole pipeline is represented in Figure 1. The hot pixels, flicker noise, and uneven background of the original observation are removed during preprocessing by background denoising and smoothing. Then, the sub-figures are extracted from the processed images. The sub-figures extraction includes a threshold method, contour detection, and centroid location. The extracted sub-figures and the added simulated space debris form the training dataset. Although the training dataset contains noisy labels, mainly stars, we forward these samples to Co-correcting to train networks with noisy labels. Finally, the trained networks are evaluated on the test dataset, and conduct space debris detection in inference stage.

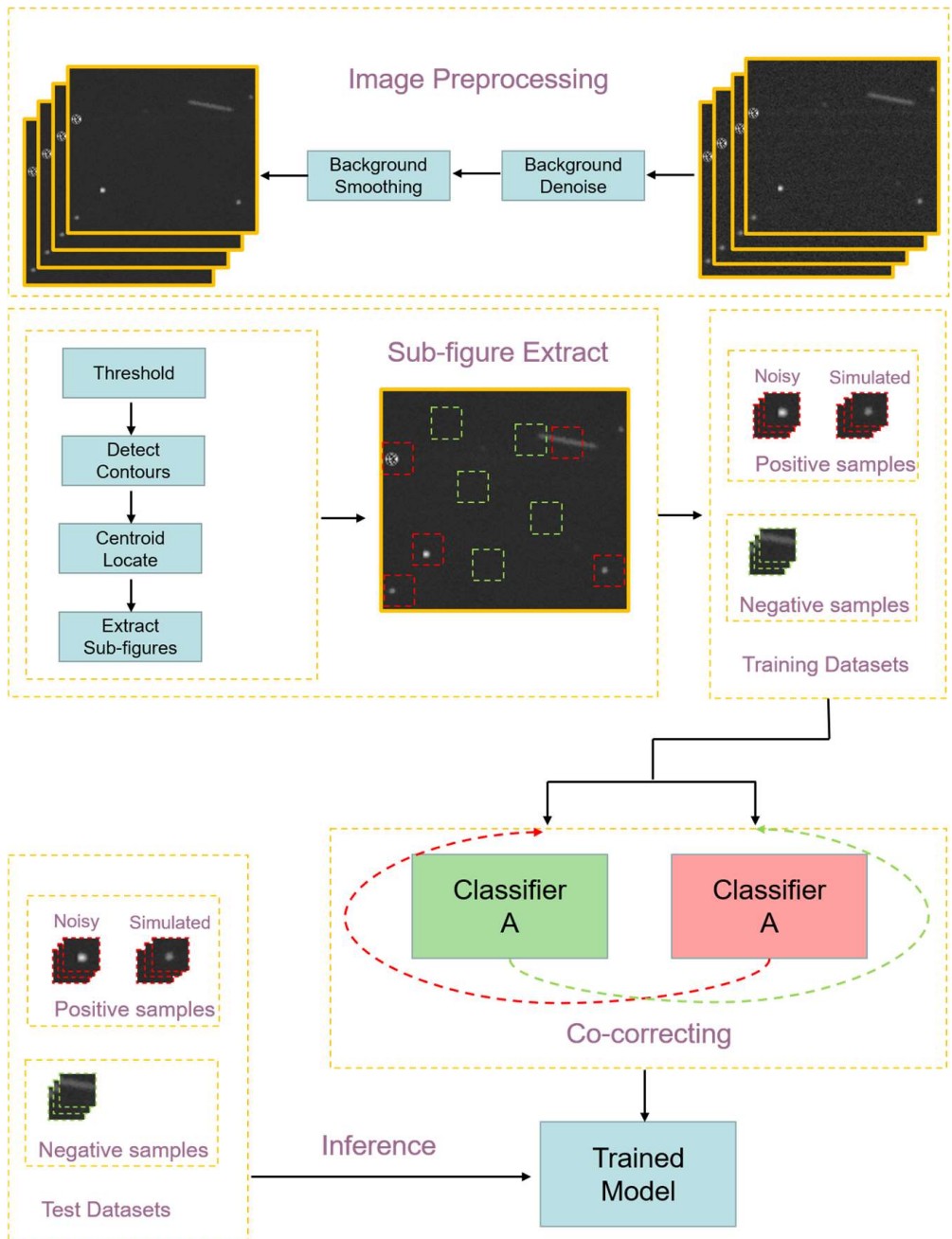

**Figure 1.** The pipeline of Co-correcting in space debris detection. The original observation images are processed by background denoise and background smoothing to acquire a clean background. Then, we extract sub-figures as training dataset. The extracted sub-figures contain plenty noisy labels. Simulated space debris is added to dataset as positive samples. Then, the dataset with noisy labels is sent to Co-correcting. The trained models by Co-correcting with noisy labels is evaluated on test dataset.

In this section, we first introduce the mathematical formulation of space debris detection and label-noise learning. Then, we thoroughly present the whole pipeline of our work, including preprocessing and the Co-correcting paradigm.

### 2.1. Problem Formulation

2.1.1. Space Debris Detection

Most communication satellites, e.g., television broadcasts, are in geostationary earth orbit (GEO), and the collisions and explosions of satellites produce a tremendous quantity of space debris. The altitude of GEO is 36,000 km, resulting in a small and dim observation of space debris. Optical telescopes are only employed for observation in GEO due to their great sensitivity over long distances. The final optical image is captured by a charge-couple device (CCD). The optical images can be modeled as

$$f(i,j) = D(i,j) + S(i,j) + B(i,j) + n(i,j) \tag{1}$$

$D(i,j)$ and $S(i,j)$ represent space debris and stars. $B(i,j)$ represents background and $n(i,j)$ is the CCD dark current noise.

Due to the difference in light condition and CCD channel, $B(i,j)$ is uneven, and contains hot pixels and flicker noise. To obtain improved performance in space debris detection, background denoising and background smoothing are essential. The background denoising seeks to reduce hot pixels and flicker noise. Hot pixels are created by damaged CCD pixels owing to cosmic radiation. Flicker noise is a single bright spot with only a few pixels in the image. Hot pixels and flicker noise can be reduced via bilateral filters [42]. Background smoothing seeks to eliminate the uneven background with mathematical morphology operators.

Debris in GEO remains relative stationary to ground-based telescopes. However, stars remain relatively stationary to earth due to their infinite distance. In "staring target mode", the telescopes remain fixed and in one direction during exposure time. The space debris keep stable in field of telescopes, and appears point-like. However, stars keep moving with specific motion in the view field of telescopes, due to rotation of the earth. If exposure time is sufficient, the trails of stars appear, and stars appear streak-like. In short time exposure, the trails of stars are subtle, and stars become similar to debris. If the mode of telescopes is set to "staring star mode", the telescope moves synchronously with the star background during the exposure. In the field of telescopes, stars remain stable, and space debris moves with expected motion. The space debris will appear in trails if the exposure time is sufficient.

In short time exposure, $D(i,j)$ and $S(i,j)$ share similar optical features, and $S(i,j)$ becomes the main interference in space debris detection. It is hard to completely remove $S(i,j)$ from $f(i,j)$. Classical methods try to remove the stars through inter-frame difference before detecting debris. However, the stars keep flickering in successive frames and are hard to remove completely. CNN is adopted to detect space debris from the background, but its performance relies on a large-scale dataset with highly accurate annotation. The annotation is extremely time-consuming due to the similarity between debris and stars; therefore, we introduce label-noise learning to avoid label annotating and data cleaning.

2.1.2. Label-Noise Learning

In space debris detection, the issue can be stated as two classifiers with output of $[0, 1]$. "1" signifies the space debris and "0" denotes the background. CNN-based methods assume the dataset $\mathcal{D}$ is obtained from clean distribution $p(X, Y)$. $X$ refers to the samples and $Y$ is the corresponding labels. However, in label-noise learning context, noisy dataset $\widetilde{\mathcal{D}}$ is derived from a corrupted distribution $p(X, \widetilde{Y})$, and $\widetilde{Y}$ is the noisy label. Assume that $N$ samples are taken from original observation images and the noisy dataset is denoted as $\widetilde{\mathcal{D}} = \{(x_i, \widetilde{y}_i)\}_{i=1}^{N}$. $x_i$ is $i$-th observed sample, and $\widetilde{y}_i \in \{0, 1\}$ is the corresponding noisy label.

Let $f(\cdot)$ indicate the (Bayes) optical hypothesis from $x$ to $y$ in clean data distribution $p(X, Y)$. In hypothesis space $\mathcal{H}$, the $f(\cdot)$ can be parameterized by $\theta$ and signified as $f_\theta(\cdot)$. The purpose is to search for optimal $f_\theta(\cdot)$ in hypothesis space $\mathcal{H}$. In label-noise learning, the $f(\cdot)$ can be parameterized by $\theta^*$, and the $f_{\theta^*}(\cdot)$ denotes the hypothesis from

$x$ to $\widetilde{y}$ in noisy distribution $p(X, \widetilde{Y})$. $f_\theta(\cdot)$ and $f_{\theta^*}(\cdot)$ can be implemented by CNN. In hypothesis space $\mathcal{H}$, the noisy distribution $p(X, \widetilde{Y})$ is expected to approximate the clean data distribution $p(X, Y)$, and therefore the label-noise learning can be redefined as that the $f_{\theta^*}(\cdot)$ is anticipated to approximate $f_\theta(\cdot)$ in hypothesis space $\mathcal{H}$.

In space debris detection, debris are typically confused with stars owing to their similarity, and the data of debris contain numerous noisy labels. Label-noise learning tries to find the optimal hypothesis $f_{\theta^*}(\cdot)$ in noisy data, and the $f_{\theta^*}(\cdot)$ is expected to have equivalent abilities to $f_\theta(\cdot)$ in clean data.

*2.2. Preprocessing*

In original observation images recorded by ground-based telescopes, the background is generally uneven, owing to variable light condition, channels of CCD, and thin clouds. The background has hot pixels and flicker noise as well. These characteristics make it harder to detect space debris in the original observation. The technique of preprocessing is depicted in Figure 1. First, we employ a bilateral filter to reduce hot pixels and flicker noise. The mathematical morphology operator is then adopted to remove the uneven background. Lastly, the processed images can be utilized to extract sub-figures of space debris by threshold and contour detection.

2.2.1. Background Denoising

In the original observation, the background contains hot pixels and flicker noise. Hot pixels and flicker noise are a single bright area with a few pixels and domain the entire image, making it difficult to distinguish objects. Since space debris is tiny and dark with very low SNR, the averaging filter and median filter easily result in object loss. To tackle this issue, we utilize a bilateral filter to eliminate hot pixels and flicker noise in the background. The bilateral filter can be defined as in Equation (2).

$$I_{filter}(x) = \frac{1}{\mathcal{W}_p} \sum_{x_i \in \Omega} k_s(||x_i - x||) I(x_i) k_r(||I(x_i) - I(x)||) \tag{2}$$

The normalization term $\mathcal{W}_p$ is defined as follows.

$$\mathcal{W}_p = \sum_{x_i \in \Omega} k_s(||x_i - x||) k_r(||I(x_i) - I(x)||) \tag{3}$$

$I_{filter}(x)$ and $I$ denote the filtered images and original input images, respectively. $x$ is the current pixels. $\Omega$ is the neighboring of $x$, and so $x_i \in \Omega$ denotes another pixel except $x$ in $\Omega$. $k_r$ and $k_s$ are the intensity kernel and the spatial kernel, respectively. $k_r$ and $k_s$ can be the Gaussian function.

For each pixel, the intensity is computed by the average of the surrounding pixels. The weight $\mathcal{W}_p$ utilize spatial kernel $k_s$ and intensity kernel $k_r$ to represent spatial closeness and intensity difference. For example, a pixel $x$ is located at $(i, j)$. The averaged intensity of $x$ is calculated on the neighboring pixels to denoise the image. $x_i$, located at $(k, l)$, denotes the neighboring pixels of $x$. Assume that $k_r$ and $k_s$ are the Gaussian kernel; the weight $w(i, j, k, l)$ is calculated by

$$w(i, j, k, l) = \exp\left(-\frac{||I(i, j) - I(k, l)||}{2\sigma_r^2} - \frac{(i-k)^2 + (j-l)^2}{2\sigma_d^2}\right), \tag{4}$$

where $I(i, j)$ and $I(k, l)$ are the intensity of the corresponding pixels; $\sigma_d$ and $\sigma_r$ are the standard deviation of spatial closeness and intensity difference, respectively. Then, normalize the neighboring by

$$I_D(i, j) = \frac{\sum_{k,l} I(k, l) w(i, j, k, l)}{\sum_{k,l} w(i, j, k, l)} \tag{5}$$

$I_D$ is the intensity of pixel $x$ from the denoised image. The results of background denoise are shown in Figure 2. We can see that the pixels and flicker noise have been filtered effectively.

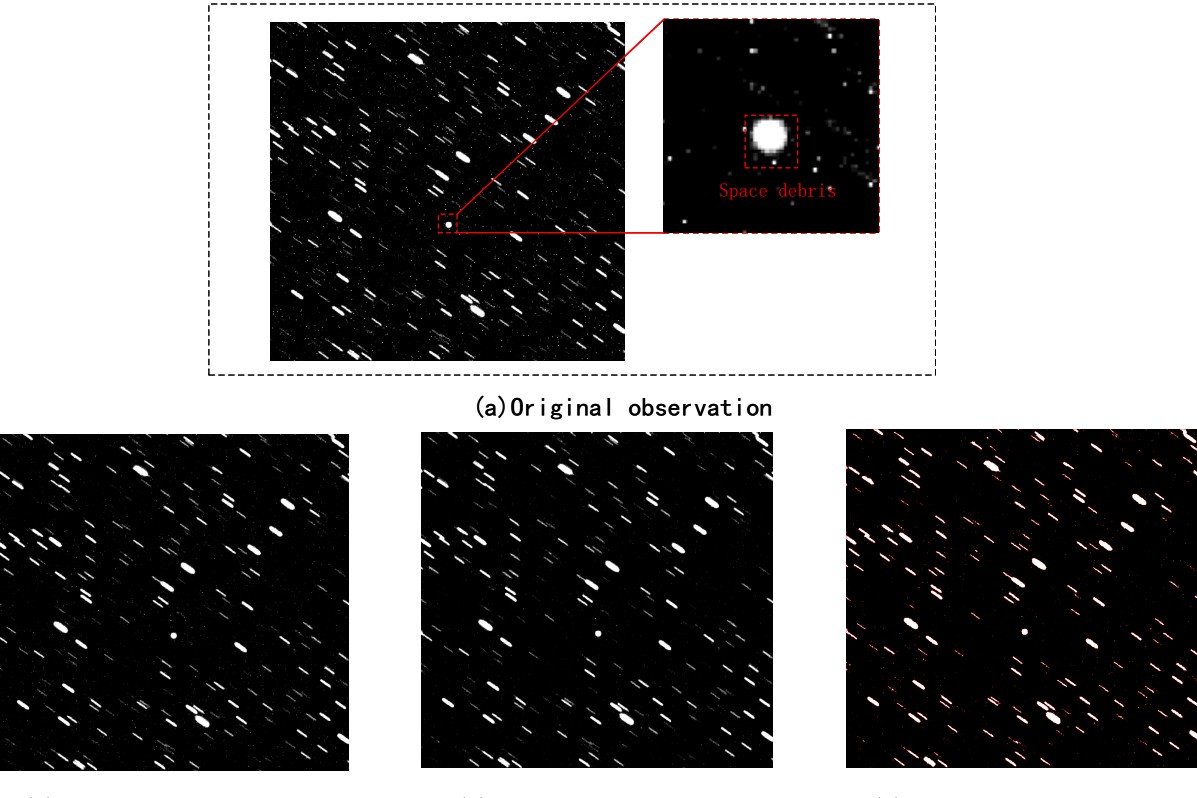

(a)Original observation

(b)Background denoising　　　(c)Background smothing　　　(d)Threshold and locating

**Figure 2.** The image is a subsection of real observation of a ground-based telescope. (**a**) The original observation image. Space debris is point-like and stars are stripe-like. (**b**)The result of background denoising. (**c**) The result of background smoothing. (**d**) The result of threshold and locating.

2.2.2. Background Smoothing

Due to the skylight condition, thin cloud and different channels of CCD, the background of the original observation is uneven. The unevenness makes it difficult to detect objects in the image. Thus, we adopt the mathematical morphology operator to smooth the uneven background.

Mathematical morphology transform consists of dilation operator and erosion operator. Let $f(i,j)$ be a reference image and $e(x,y)$ be the structuring element. The dilation operator is defined as

$$(f \oplus e) = \min\left\{f(i-x,j-y) + e(x,y)|(i-x,j-y) \in D_f, (x,y) \in D_e\right\}, \tag{6}$$

the minimum value of pixels are assigned to the image border.

The erosion operator is defined as

$$(f \ominus e) = \max\left\{f(i+x,j+y) - e(x,y)|(i+x,j+y) \in D_f, (x,y) \in D_e\right\}, \tag{7}$$

the maximum value of pixels are assigned to the image border.

The structuring element $e(x,y)$ plays a great role in dilation and erosion operators. A structuring element is a matrix consisting of only zero and one that can have any arbitrary shape and size. Structuring elements are determined by the size of space debris and stars, the view field of telescopes and the pattern of observation. If the size of the structuring element is too tiny, the stars and space debris will be eliminated by mistakes. Instead, the unevenness of the background cannot be successfully smoothed if the size of the structuring

element is too large. In this paper, the size of structuring element is fixed at $5 \times 5$. We initially perform the erosion operator and then conduct the dilation operator. The image with an even background is obtained by

$$f_{smooth} = (f \ominus b) \oplus b, \tag{8}$$

$f_{smooth}$ is the smoothing image of original observation with uneven background.

### 2.2.3. Sub-Figure Extraction

After background denoising and smoothing, the sub-figure of space debris can be extracted. Firstly, we perform threshold methods on smoothing images, and then detect the contours of the object in images to extract sub-figure.

The results of threshold can be defined as follows:

$$f_{threshold}(i,j) = \begin{cases} 1, & if\ f(i,j) \geq threshold \\ 0, & if\ f(i,j) \leq threshold \end{cases} \tag{9}$$

And the threshold is set as:

$$threshold = m + k\delta, \tag{10}$$

where $m$ and $\delta$ are the mean and standard deviation of the background, respectively, and $k$ is the coefficient determined by the number of points.

Then, we detect the contours of objects in image $f_{threshold}$, and each contour corresponds to one object. The centroid of each object is calculated by contours, as follows.

$$
\begin{aligned}
x_0 &= \frac{\sum_{k=1}^{n} x_k}{n} \\
y_0 &= \frac{\sum_{k=1}^{n} y_k}{n}
\end{aligned}
\tag{11}
$$

$n$ is the total number of pixels in every contour, and $(x_0, y_0)$ is the coordinates of centroid. Then, we clip a $32 \times 32$ sub-figure from $f_{smooth}$ as positive samples for the following training. The negative samples are randomly selected from $f_{smooth}$ with $32 \times 32$ patches. The result of threshold method and locating can be seen in Figure 2.

### 2.3. Co-Correcting

In space debris detection, the samples directly derived from images commonly include noisy labels, mainly due to the low SNR and similarity to stars. To avoid time-consuming procedures such as data cleaning, we directly employ the samples with noisy labels as a compromise to train networks by introducing the methodology of label-noise learning. We propose a novel label-noise learning paradigm, Co-correcting, to correct the noisy labels and update parameters with the corrected labels. Concretely, we randomly initialize two identical networks and then make their own predictions on the same samples. The predictions of each network serve as auxiliary supervised information to correct the noisy labels for peer networks during parameters updating stage. In other words, we utilize the predictions of each network to correct its peer network's original targets of noisy labels. Due to the memorization effects of DNN, the networks have the ability to recognize the noisy samples at the beginning of training. The corrected labels provide more supervised information which can suppress the negative impact of noisy labels effectively. Thus, our proposed method is termed "Co-correcting". The supervision of our proposed Co-correcting can be separated into two parts: original targets and auxiliary supervised information. Original targets derive from the noisy datasets (i.e., space debris directly extracted form images) and will degenerate the performance of networks. Consequently, we utilize the auxiliary supervised information from the peer networks' predictions to correct the original targets. By the mutual rectification, Co-correcting has stronger supervision to prevent overfitting noisy labels.

In a real engineering context, we train two identical networks A and B, denoted by $f(x, \theta_1)$ and $f(x, \theta_2)$. $\theta_1$ and $\theta_2$ are parameters of networks A and B. $p_1^i \in \{0, 1\}$ and $p_2^i \in \{0, 1\}$ are their own predictions of $i$-th samples, respectively.

### 2.3.1. The Structure of Co-Correcting

The structure of Co-correcting is depicted in Figure 3. In Co-correcting, two identical networks are utilized to make predictions on the same mini-batch data. We argue that each network can provide effective supervised information for its peer networks. Due to random initialization, two networks with the same structure have different abilities and perspectives towards the same mini-batch data. It is the divergence that is vital to our proposed method. For the same mini-batch data, network A and network B have their respective predictions. These predictions depend on the parameters of two networks that are updated during Stochastic Gradient Descent (SGD) [43]. Distinct predictions represent different perspectives of two networks towards the same data. Based on this view, two networks can learn from each other. In label-noise learning, the labels cannot provide absolutely correct supervision owing to the existence of noise. To address this issue, we use the predictions of two networks as auxiliary supervised information. The memorization effect of DNN can be divided into two steps. At the first step, networks tend to memorize samples with clean labels and then gradually overfit towards hard samples with noisy labels at the second stage. Hence, the auxiliary supervised information from peer networks can be exploited in the second training step to prevent overfitting noisy samples.

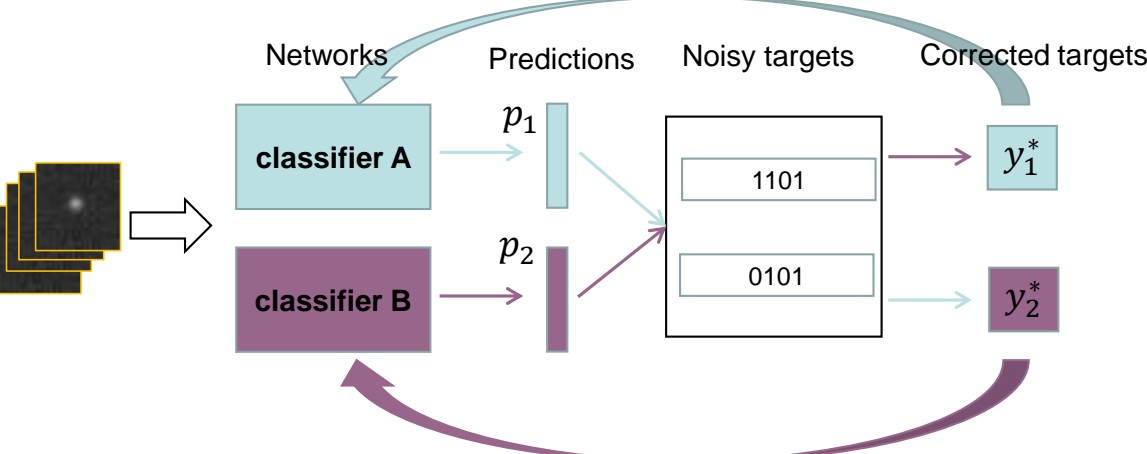

**Figure 3.** The schematic of Co-correcting. In Co-correcting, two networks make predictions on same mini-batch data, and the noisy labels are corrected by peer networks' predictions. The new corrected labels then guide the training of the networks. We can see that each network is under supervision of its peer network.

To implement this, we introduce the new target $y^*$ corrected by peer networks' predictions and compute loss function on $y^*$.

$$
\begin{aligned}
y_1^* &= \alpha y + (1 - \alpha) p_2 \\
y_2^* &= \alpha y + (1 - \alpha) p_1
\end{aligned}
\tag{12}
$$

$y_1^*$, $y_2^*$ are the new corrected targets for classifier $f(x, \theta_1)$ and $f(x, \theta_2)$. $y$ is the original target of noisy labels and $\alpha$ controls the extent of correction from peer networks. $y_1^*$ consists of two parts: noisy labels $y$ as the original supervised target, and the peer network's predictions as the new supervised target. Due to the existence of noise, we provide $p_2$ as extra supervised information to correct noisy labels $y$. Intuitively, the networks are more robust, the auxiliary supervised information is more effective, and it even approximates the case in clean labels learning.

The memorization effect of deep networks reveals that training of DNN can split into two phases. Based on this observation, we propose a small $\alpha$ in Equation (12) in early training, and then the $\alpha$ increases to noise rate $\varepsilon$. At the beginning of training, networks have formed their own perspectives and can select reliable samples from noisy dataset. Due to this, the clean labels dominate the whole training data. Here, supervised information from peer networks is not essential for training. Yet, as the epoch increases, side effects of noisy labels tend to appear. The networks cannot handle noisy labels by just relying on their inherent properties learned in the first step through clean labels. Consequently, they require more vigorous supervision. A big $\alpha$ is required to improve the supervision in Equation (12). As in Equation (13), $t$ refers to the current epoch, $E_k$ regulates the pace of $\alpha$ attaining its maximum. The maximum of $\alpha$ is defined by the noisy rate $\varepsilon$.

$$\alpha(t) = \varepsilon \min\{\frac{t}{E_k}, 1\} \tag{13}$$

### 2.3.2. Loss Function of Co-Correcting

The loss function of Co-correcting is calculated by corresponding new corrected labels.

$$\begin{aligned}
\ell_1(x_i) &= CCE(p_1, y_1^*) \\
&= -\sum_{i=1}^{N} y_1^{i*} \log p_1^i(x_i) \\
\ell_2(x_i) &= CCE(p_2, y_2^*) \\
&= -\sum_{i=1}^{N} y_2^{i*} \log p_2^i(x_i)
\end{aligned} \tag{14}$$

CCE signifies Categorical Cross Entropy. We apply Cross-Entropy Loss to minimize the distance between predictions and corrected targets. The corrected targets are influenced by peer networks so the CCE can be seen to minimize the distance between two networks of Co-correcting. Each network is under the supervision of its peer network. Due to the divergence of two networks, they might present distinct viewpoints upon the same mini-batch data. Thus, each network can distill supervised information from mini-batch data to guide the training of another peer network.

During back-propagation, the error accumulation of each network caused by mini-batch data can be successfully reduced by the procedure of mutually correcting peer networks' targets.

### 2.3.3. Small-Loss Selection

Due to the memorization effect of DNN [31], we should train networks with clean samples to prevent them overfitting noisy labels. Samples with small loss are more likely to have the clean labels [30,34,40]. Co-teaching utilizes two identical networks with different initialization to select clean samples. Each network selects samples with small-loss policy on their own perspectives, and then the selected samples are utilized for their peer networks. Instead, JoCoR utilizes joint regularization to select small-loss samples upon perspectives of both networks. JoCoR argues that the samples are more likely to have clean labels when two networks achieve agreement. Similar to Co-teaching and JoCoR, we select samples of each network with small-loss policy based on Equation (15).

$$\begin{aligned}
\widetilde{\mathcal{D}}_i &= \arg\min_{\mathcal{D}_i' : |\mathcal{D}_i'| \ge R(t)|\bar{\mathcal{D}}|} \ell_i(\mathcal{D}_i') \\
i &= 1, 2
\end{aligned} \tag{15}$$

In Equation (15), $\widetilde{D}_1$ and $\widetilde{D}_2$ are selected samples of two networks, respectively. $\bar{\mathcal{D}}$ is the mini-batch dataset. Let $R(t)$ determine the number of selected samples in every

mini-batch [40]. Compared to Co-teaching, we do not employ the "cross update" approach. In our study, the training samples of two networks are the same (Co-teaching is not), and we take the union of these chosen samples as training data for both networks.

$$\widetilde{\mathcal{D}} = \widetilde{\mathcal{D}}_1 \cup \widetilde{\mathcal{D}}_2 \tag{16}$$

Then, networks forward propagate on $\widetilde{D}$ and update parameters during backward propagation. We argue that such a sample selecting policy can utilize more data and take advantage of two networks' different perspectives.

2.3.4. Algorithm Description

In Algorithm 1, $f(x, \boldsymbol{\theta}_1)$ and $f(x, \boldsymbol{\theta}_2)$, respectively, select small-loss samples from mini-batch data (step 5–6) based on their own loss function in Equation (14). Then, we take the union of two networks' selected samples as training samples (step 7). To strengthen the supervision of training, we correct the original targets of noisy labels by peer networks' predictions (step 8–9). The loss function of each network is computed on the corresponding corrected target, and then back-propagation is conducted to update two networks' parameters, respectively (step 10–11).

---

**Algorithm 1** Co-correcting

---

**Input**: Networks $f(x, \boldsymbol{\theta}_1)$ and $f(x, \boldsymbol{\theta}_2)$, training dataset $D$, learning rate $\eta$, noisy rate $\varepsilon$ and epoch $E_k$ and $E_{max}$, iteration $I_{max}$
**Output**: Networks $f(x, \boldsymbol{\theta}_1)$ and $f(x, \boldsymbol{\theta}_2)$

1: **for** $t = 1, 2, \ldots, E_{max}$ **do**
2:    **Shuffle** training set $\mathcal{D}$;                                      //noisy dataset
3:    **for** $i = 1, 2, \ldots, I_{max}$ **do**
4:       **Fetch** mini-batch $\bar{\mathcal{D}}$ from $\mathcal{D}$
5:       **Select** $\widetilde{\mathcal{D}}_1 = \arg\min_{\mathcal{D}'_1 : |\mathcal{D}'_1| \geq R(t)|\bar{\mathcal{D}}|} \ell_1(\mathcal{D}'_1)$
6:       **Select** $\widetilde{\mathcal{D}}_2 = \arg\min_{\mathcal{D}'_2 : |\mathcal{D}'_2| \geq R(t)|\bar{\mathcal{D}}|} \ell_2(\mathcal{D}'_2)$
7:       **Obtain** $\widetilde{\mathcal{D}} = \widetilde{\mathcal{D}}_1 \cup \widetilde{\mathcal{D}}_2$
8:       **Correct** $y_1^* = \alpha y + (1 - \alpha) p_2$
9:       **Correct** $y_2^* = \alpha y + (1 - \alpha) p_1$
10:      **Update** $\boldsymbol{\theta}_1 = \boldsymbol{\theta}_1 - \eta \bigtriangledown \ell_1(\widetilde{\mathcal{D}})$
11:      **Update** $\boldsymbol{\theta}_2 = \boldsymbol{\theta}_2 - \eta \bigtriangledown \ell_2(\widetilde{\mathcal{D}})$
12:    **end for**
13:    **Update** $\mathcal{R}(t) = 1 - \varepsilon \min\{\frac{t}{E_k}, 1\}$
14:    **Update** $\alpha$ based on Equation (13)
15: **end for**

---

## 3. Results

In this section, we introduce the concrete experiment setting and exhibit the concise description of the experimental results.

### 3.1. Experiments Setting

3.1.1. Dataset

In this paper, the original observation is processed by background denoising in Section 2.2.1 and background smoothing in Section 2.2.2. Then, the sub-figures are clipped from the original observation images with the size of $32 \times 32$, and serve as positive training samples for networks. Note that the extracted sub-figures of positive samples contain space debris and false alarms with noisy labels. Space debris is the positive sample and stars are the false alarms. In our work, the extracted sub-figures are identified as space debris, i.d., the sub-figures are assigned a positive label "1", despite the labels contain ambiguous

category. The negative samples with label "0" background are randomly cropped from original observation images with the size of $32 \times 32$.

In label-noise learning, the label noise rate must be lower than 50% for the two-classifier. In other words, the right labels must be prominent. In original observation images, the number of stars is significantly larger than the amount of space debris. Thus, simulated space debris is added to training dataset to expand the number of clean samples. The intensity distribution of the debris without motion blur is

$$I(x,y) = I_0 \left( \frac{2J_1 \left( \sqrt{x^2 + y^2} \right)}{\sqrt{x^2 + y^2}} \right)^2 \tag{17}$$

where $I_0$ is the central intensity of the debris, and $J_1$ is the first-order Bessel function.

In this way, despite the extracted sub-figures from images include false alarms (e.g., stars), space debris dominates the positive samples, i.e., the number of correct labels in the training dataset is larger than the wrong labels. We prepare four datasets with different noise rate.

In every image, we extract 100 sub-figures of the detected objects (most of them are stars and a few are space objects) in Section 2.2.3, and then 100/200/300/500 simulated space debris is added to each image. The 100 extracted patches and simulated space debris serve as positive samples. As for negative samples, we randomly extract 200/300/400/600 patches from the background in every image to balance the quantity of positive and negative samples. We prepare 10 original observation images with size of $4096 \times 4096$ for training, and the total number of training samples is 4000/6000/8000/12,000 with corresponding estimated label noise rate at 50%/33.3%/25%/16%. The samples in the test dataset share the same distribution with training data, and the test dataset has 300 positive samples and 300 negative samples extracted from the other three original observation images.

### 3.1.2. Baselines

Our proposed Co-correcting (Algorithm 1) is compared with the following state-of-the-art methods. For fair comparison, we implement all methods in **Baselines** with default parameters and same network architectures. All methods are implemented by PyTorch, and all experiments are conducted on NVIDIA RTX 3090 GPU.

- JoCoR [30], which trains two networks and adopts joint training with Co-regularization to combat noisy labels.
- Co-teaching [40], which trains two networks simultaneously and cross-updates parameters of peer networks.
- Standard, which trains networks directly on noisy datasets as a simple baseline.

### 3.1.3. Measurement

We use test accuracy to measure the performance of every method. Test accuracy is defined as follows.

$$test\ accuracy = \frac{the\ num.\ of\ correct\ predictions}{the\ num.\ of\ test} \tag{18}$$

Besides, label precision i.e.,

$$label\ precision = \frac{the\ num.\ of\ clean\ labels}{the\ num.\ of\ all\ selected\ labels} \tag{19}$$

is also calculated. Label precision reflects the abilities of methods in select reliable samples. The higher label precision means more clean samples are selected to train networks, and thus the networks may achieve high performance in test accuracy.

We also introduce detection probability and false alarm rate to measure the performance of space debris detection. They are given by

$$P_d = \frac{N_d}{N_{total}}$$
$$P_f = \frac{N_f}{N_{total}}$$

(20)

where $P_d$ is detection probability and $P_f$ is false alarm rate. $N_d$, $N_f$ and $N_{total}$ represents numbers of detected space debris, false alarms, and total space debris.

We implement all experiments five times. The highlighted shade in each figure denotes the standard deviation.

### 3.1.4. Network Structure and Optimizer

Because space debris only covers a few pixels and has low-level features such as edges and points, the shallow neural networks are suitable to this situation. Specifically, we adopt the two-layer CNN architecture in Table 1 as backbone networks for all methods in **Baselines**. For all experiments, active function is ReLU and batch-size is set to 64. We adopt an Adam optimizer with momentum 0.9 and set the initial learning rate to 0.001. Dropout and bath normalization are also used. We run 50 epochs for all experiments.

**Table 1.** The structure of 2-layer CNN.

| 2-layer CNN |
| :---: |
| $32 \times 32$ gray Image |
| $5 \times 5$, 16 BN, ReLU |
| $2 \times 2$ Max-pool |
| $5 \times 5$, 16 BN, ReLU |
| $2 \times 2$ Max-pool |
| Dense $16 \times 5 \times 5 \longrightarrow 120$, ReLU |
| Dense $120 \longrightarrow 84$, ReLU |
| Dense $84 \longrightarrow 10$ |

### 3.1.5. Selection Setting

The noisy rate $\varepsilon$ can be inferred by the training dataset. For example, the positive samples in an image have $200/300/400/600$ simulated space debris, and the corresponding noisy rate $\varepsilon$ is about $50\%/33.3\%/25\%/16\%$, respectively. The ratio of small-loss samples $R(t)$ of Co-correcting is set to:

$$R(t) = 1 - \min \varepsilon \left\{ \frac{t}{E_k}, 1 \right\}$$

(21)

$R(t)$ determines the selected sample ratio in small-loss selection. $t$ is the current epoch and $E_k$ is the scheduled epoch, where Co-correcting selects most samples in every updating procedure.

### 3.2. Feasibility of Label-Noise Learning in Space Debris Detection

In this section, we demonstrate the feasibility of label-noise learning methods in space debris detection. All methods in **Baseline** are implemented with the same settings in Section 3.1.4. We train all networks on datasets with four different noise rates: 50%/ 33.3%/ 25%/ 16%, and then trained networks are tested on the test dataset.

As in Figure 4, the performance of each method on a noisy dataset with a different noise rate is plotted. We can see the memorization effect clearly in all four plots. The standard test accuracy rapidly achieves the greatest levels at the first stage, and at the second stage it declines gradually owing to existence of noise labels. However, we can see that the other methods can successfully alleviate the decline at the second stage. At this point, all methods demonstrate their effectiveness in avoiding networks overfitting

towards noisy labels. To illustrate these phenomena, we plot the label precision curve in Figure 5 of every method in all cases with a different noise rate. We can observe that the label precision of every method has the same trend compared to test accuracy. The label precision rises initially and then falls progressively. The rationale is that the label-noise learning methods can extract clean labels from noisy datasets by themselves. Test accuracy reaches its maximum in the seventh epoch, but label precision reaches its highest at the 10th epoch. The delay in the label precision's diminishing illustrates that the networks begin to overfit the noisy data, and then lose the ability to select clean labels. The selected noisy labels degenerate the performance of networks in turn. The detailed test accuracy can be seen in Table 2, and the detailed label precision can be seen in Table 3.

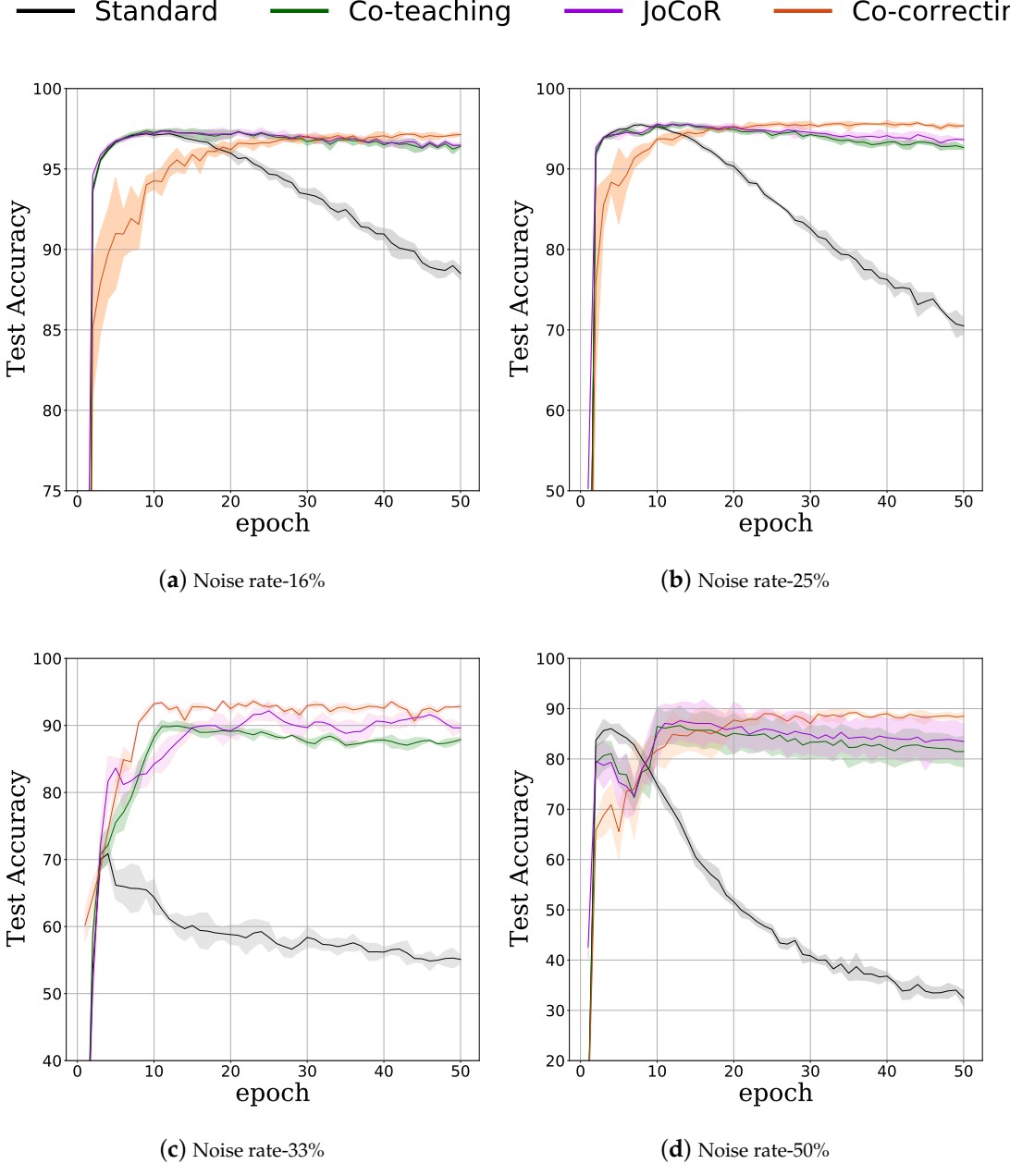

**Figure 4.** The test accuracy of the methods in **Baselines**. The datasets have the different label noise rate. The label noise rate is: (**a**) noise rate 16%, (**b**) noise rate 25%, (**c**) noise rate 33%, and (**d**) noise rate 50%. Each experiment is repeated five times. The error bar for STD in each figure has been highlighted as a shade.

**Table 2.** The test accuracy of methods in **Baselines** with different noise rate.

| Noise rate | Standard | Co-Teaching | JoCoR | Co-Correcting |
|---|---|---|---|---|
| Noise rate 16 % | 89.34 ± 0.50 | 96.46 ± 0.22 | 96.67 ± 0.20 | **97.55** ± 0.07 |
| Noise rate 25 % | 73.14 ± 0.90 | 93.04 ± 0.36 | 94.49 ± 0.53 | **95.95** ± 0.11 |
| Noise rate 33 % | 55.52 ± 0.98 | 97.48 ± 0.56 | 90.67 ± 0.73 | **92.45** ± 0.45 |
| Noise rate 50 % | 33.97 ± 1.32 | 82.12 ± 3.02 | 85.49 ± 4.03 | **88.45** ± 0.75 |

The test accuracy of every method is calculated over the last 10 epochs.

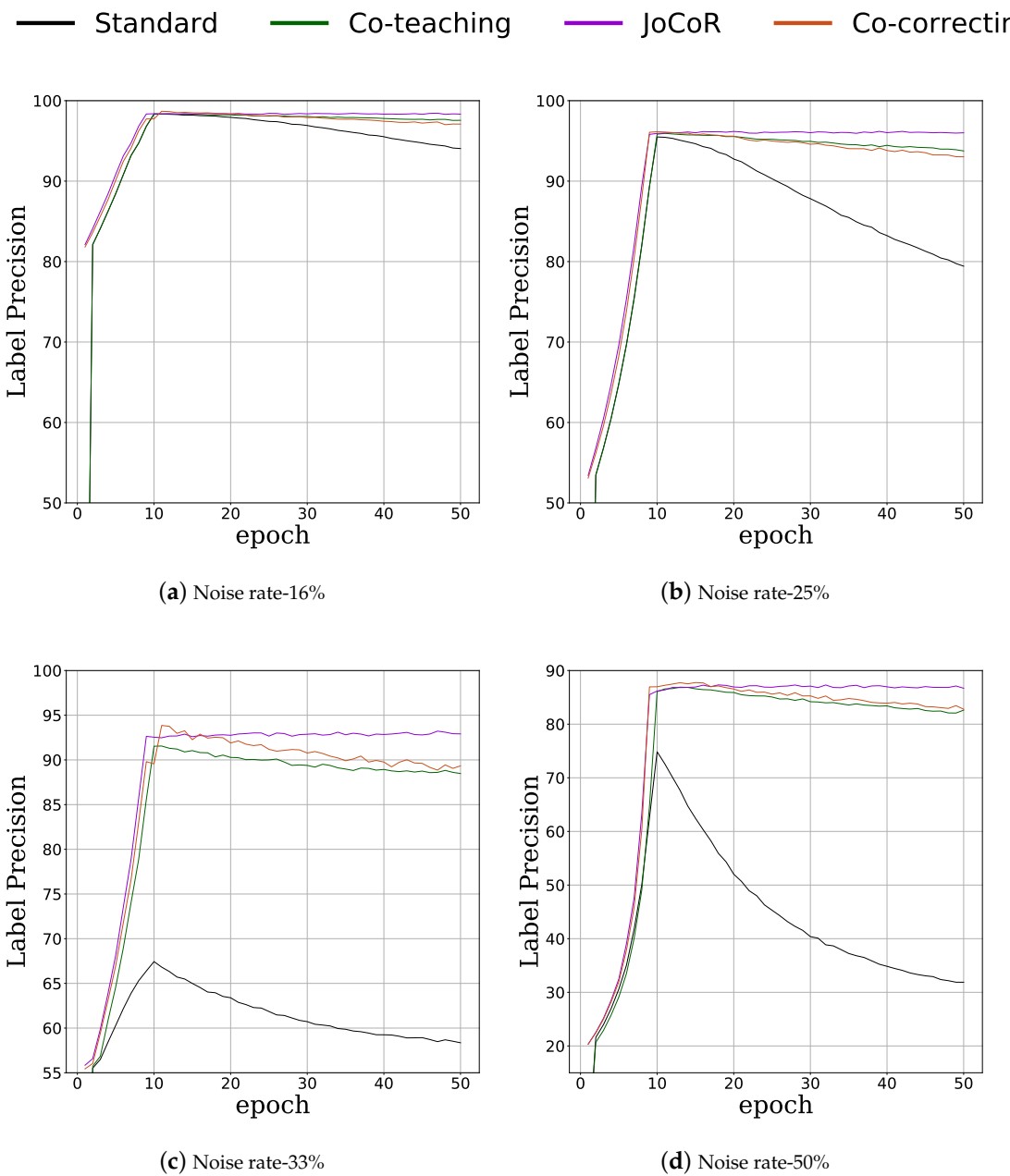

**Figure 5.** The label precision of the methods in **Baselines**. The training datasets have four different label noise rate: (**a**) noise rate 16%, (**b**) noise rate 25%, (**c**) noise rate 33% and (**d**) noise rate 50%. Higher label precision means less noisy samples are selected during sample selection, and methods with high label precision are more robust to noisy labels.

**Table 3.** The label precision of methods in **Baselines** with different noise rate.

| Noise Rate | Standard | Co-Teaching | JoCoR | Co-Correcting |
|---|---|---|---|---|
| Noise rate 16 % | 94.67 | 97.66 | **98.36** | 97.24 |
| Noise rate 25 % | 81.10 | 94.09 | **96.05** | 93.41 |
| Noise rate 33 % | 58.78 | 88.67 | **92.97** | 89.41 |
| Noise rate 50 % | 32.98 | 82.59 | **86.87** | 83.43 |

The label precision of every method is calculated over the last 10 epochs.

When the noise rate is 16%, the test accuracy of Standard starts to fall after it achieves the greatest level at the 7th epoch. Finally, the Standard achieves the test accuracy at 89.34%. However, other methods such as Co-teaching, JoCoR and Co-correcting work well; notably, our proposed Co-correcting achieves the best result at 97.55%. It is noticed that in the 17 early epochs, the test accuracy of Co-correcting is lower than Co-teaching and JoCoR, and then other methods begin to decline, but Co-correcting grows progressively. This is because Co-correcting can correct the noisy labels via its two identical peer networks. Although Co-teaching and JoCoR begin to drop significantly, Co-correcting can still grow in test accuracy. In label precision, all approaches yield great results due to the low noise rate. The JoCoR obtains the best performance in label precision, while Co-correcting still surpasses other methods in test accuracy.

In the case of 25% noise rate, the tendency is same as the case of 16% noise rate. The Co-correcting achieves the best test accuracy in the 30 late epochs at $95.95 \pm 0.11\%$, but JoCoR achieves the maximum label precision.

In the case of 33% noise rate, the Standard degenerates substantially and ultimately reaches $55.52 \pm 0.98\%$ test accuracy. Co-teaching, JoCoR and Co-correcting still perform well, and Co-correcting is better than other methods. JoCoR also have the performance on picking clean labels.

In the harshest case of 50% noise rate, the Standard loses its capacity to recognize objects. The test accuracy is $33.97 \pm 1.32\%$ and the label precision is 32.98%. Co-teaching, JoCoR, and Co-correcting attain the lowest test accuracy at the 7th epoch and then grow fast. In the late epochs, they maintain the high test accuracy. The trend of test accuracy is similar to label precision.

Co-correcting gets the greatest test accuracy in all cases, but JoCoR acquires the best performance of label precision. This result explains that Co-correcting can successfully correct the noisy labels using their two identical peer networks. In contrast, Co-teaching and JoCoR strive to pick clean labels, and hence their label precision is better, but their test accuracy is lower than that of Co-correcting.

### 3.3. Detection Results of Co-Correcting

In this section, we exhibit the results of space debris detection in different noise rates. The networks of Co-correcting are trained with datasets of four different noise rates: 16%, 25%, 33%, and 50%. The four trained networks are evaluated on 300 space debris samples (100 samples from 10 real images and 200 simulated samples). The detection results are shown in Table 4. When the noise rate is low (16%), Co-correcting can obtain 99.7% detection probability and 0.3% false alarms rate. Even in the harshest case (50% noise rate), the detection probability is 98.0% and the false alarm rate is 2.0%.

Due to the similarity between space debris and stars, the stars are the main source of false alarms. In training dataset, there are numerous noisy labels which provide networks with the wrong supervised information. However, our proposed Co-correcting can effectively correct the noisy labels by their peer networks mutually. The trained networks have the great performance in space debris detection with high detection probability and low false alarm rate. The results demonstrate the excellent performance of the networks trained by Co-correcting in space debris detection, even in the high noise rate case (50% noise rate).

**Table 4.** Detection results of the networks trained by Co-correcting with different noise accuracy.

| Noise Rate (%) | Total Number | Detection Number | Detection Probability (%) | False Alarms | False Alarms Rate (%) |
|---|---|---|---|---|---|
| 16% | 300 | 299 | 99.7% | 1 | 0.3% |
| 25% | 300 | 298 | 99.3% | 2 | 0.7% |
| 33% | 300 | 297 | 99.0% | 3 | 1.0% |
| 50% | 300 | 294 | 98.0% | 6 | 2.0% |

The networks of Co-correcting are trained using a dataset with a different noise rate. The 300 samples for test contain no noisy label.

## 4. Discussion

### 4.1. The Memorization Effect of Network

In Figure 4, we can see the memorization effect clearly. Standard is just a two-layer network without any label-noise learning methodology. The samples with noisy labels are directly forwarded to Standard, and its performance can reveal the memorization effect of CNN. The test accuracy of Standard increases quickly and achieves the highest level at the 8th epoch. Then, the test accuracy begins to fall. The degeneration of networks' performance in later epochs is caused by the noisy labels. In space debris detection, the training data contains numerous noisy labels, due to the similarity between debris and stars and low SNR. The parameters of networks are initialized randomly, and the number of clean labels dominates the training dataset. Thus, the networks can select the correct space debris at the first stage. With the training epoch increasing, the networks obtain more noisy labels to guide their training and thus converge to noisy labels. In Figure 5, Standard has high label precision at the first 10 epochs. That is to say, Standard can select the clean labels by itself. Then, most training samples of Standard have noisy labels during the late epochs, and the noisy labels degenerate the performance of Standard in terms of test accuracy.

### 4.2. The Feasibility of Label-Noise Learning in Label-Noise Learning

In Figure 4, Co-correcting outperforms other methods in **Baseline** in test accuracy. Co-teaching and JoCoR mainly focus on selecting clean labels during parameter updating. Instead, our proposed Co-correcting aims to correct the noisy labels. Thus, in the case of 16% and 25% noise rate, we can see that the test accuracy of Co-teaching is lower than that of Co-teaching and JoCoR at an early stage, but then it begins to increase and finally achieves the highest level. In Figure 5 of label precision, although JoCoR achieves the best performance in selecting clean labels, Co-correcting can achieve the best test accuracy. These phenomena demonstrate the effectiveness of the correcting strategies of Co-correcting. The inputted samples with noisy labels can be corrected by the two peer networks of Co-correcting. The corrected labels serve as new supervised information for networks to guide the training procedure.

### 4.3. The Performance of Co-Correcting in Space Debris Detection

The detection results of Co-correcting are shown in Table 4. In short time exposure, debris extracted from observation contains lots of false alarms, which are the main source of noisy labels in datasets. The network trained by Co-correcting with these noisy labels can achieve high detection probability and a low false alarm rate. Our work provides a new paradigm in space debris detection. In short time exposure, debris and stars share similar features. CNN can effectively detect space debris from the background, but the large datasets with highly accurate annotation are required to train networks. In space debris detection, the datasets contain lots of noisy labels. The data cleaning is costly and time consuming; therefore, Co-correcting utilizes the samples with noisy labels to train networks. Assuming that the clean labels dominate, i.e., the highest label noise rate of a two-classifier is 50%, the networks trained by Co-correcting can accomplish the space debris detection tasks with high detection probability.

## 5. Conclusions

In this paper, we propose a novel learning paradigm, Co-correcting, to overcome the noisy label in space debris detection. In original observation images, space debris contains plenty of noisy labels, making it difficult to train networks.

Our proposed Co-correcting comprises two identical networks and can correct the noisy labels with the peer networks' predictions. Empirical experiments show that Co-correcting outperforms other state-of-the-art methods of label-noise learning in space debris detection. When the label noise rate is 16%/ 25%/ 33.3%/ 50%, Co-correcting achieves the best test accuracy at 97.55%/ 95.95%/ 92.45%/ 88.45%. Even with a high label noise rate (50%), the networks trained via Co-correcting can detect space debris with a high detection probability at 98.0% and a low false alarm rate at 2.0%. These results show that Co-correcting can effectively mitigate the effects of noisy labels in space debris. In future work, we will investigate more applications of Co-correcting in space debris, such as space debris tracking, background suppression, and image registration.

**Author Contributions:** Conceptualization, H.L. and Z.N.; methodology, H.L.; software, H.L.; validation; formal analysis, H.L.; investigation, H.L.; resources, Z.N., Y.L. and Q.S.; writing—original draft preparation, H.L.; writing—review and editing, Z.N. and Y.L.; visualization, H.L.; supervision, Z.N.; project administration, Z.N.; funding acquisition, Z.N. All authors have read and agreed to the published version of the manuscript.

**Funding:** This research was funded by the Youth Science Foundation of China (Grant No.61605243).

**Data Availability Statement:** The data used to support the findings of this study are available from the corresponding author upon request

**Acknowledgments:** The authors would like to thank all the reviewers for their valuable contributions to our work.

**Conflicts of Interest:** The authors declare no conflicts of interest.

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
