# Peer review of "Co-Correcting: Combat Noisy Labels in Space Debris Detection"

_remotesensing, doi:10.3390/rs14205261_

Round 1
Reviewer 1 Report (Previous Reviewer 4)
In this paper, the authors presented a deep learning paradigm called "co-correcting" that was inspired from "co-teaching" presented in reference [18]. it was evaluated in the context of debris detection from remote sensing images.
General comments:
The manuscript presents some English writing errors:
reference[4] should be at the end of the sentence in line 42.
In line 71, the stars remain.
In line 98, these methods need
In line 107, to remove
In line 297, relatively stationary
In line 368, weight
There other errors that authors should carefully check.
Here are some of the questions from the previous round as I have not seen the responses from the authors.
The co-correcting as proposed is simply based on interchanging p1 and p2 in the loss functions. So why it is supposed to be a co-correcting and not a co-erroring ? there is no guarantee that the predicted p1 is correct nor p2. What is the rationale behind this?
The problem will occur in the beginning of the training, when \alpha is small and p_i is not the expected class.
With t> Ek , the co-correcting should have no effect on the training as per equation 12, and should converge to a Standard method by definition. But this is not the case in figure 4. Is there any explanation, if this is the case?
In equation 13, it is not clear that "t" refers to the current epoch as defined later in algorithm 1.
Since we have smaller patches (images), did the authors try more larger batches with respect to R(t) ?
In the prediction phase, which model is used, A, B or both? Is it possible to evaluate separately the performances of A and B?
Author Response
Response to Reviewer 1 Comments
Thanks for your valuable comments, the following is my response to your questions.
All revision about your valuable comments are in red in manuscript.
Point 1: The co-correcting as proposed is simply based on interchanging p1 and p2 in the loss functions. So why it is supposed to be a co-correcting and not a co-erroring ? there is no guarantee that the predicted p1 is correct nor p2. What is the rationale behind this?
Response 1: In memorization effects of DNN, we can know that DNN can recognize the clean data by itself, due to the clean data is dominating in dataset. But with the increasing of epochs, more noisy data are inputted to DNN, DNN begins to overfit noisy data. Based on this point, the predictions of DNN in early epoch is reliable and these predictions can serve as relative reliable supervised information for the peer networks. Thus, our proposed method is called Co-correcting.
DNN have great ability in recognize the true catalog at early stage due to clean labels dominate. At this time, p1 and p2 are reliable, and can be used as auxiliary information to supervise training. We add this description in line 419-428
Point 2: The problem will occur in the beginning of the training, when \alpha is small and p_i is not the expected class.
Response 2: At early training, the networks can recognize most clean data, due to the clean data are dominate. At this stage, we adopt a small \alpha, because we want the two networks to be different, in other words, have the different perspectives. If the two networks are the same, the interchanging becomes unnecessary.
Point 3: With t> Ek , the co-correcting should have no effect on the training as per equation 12, and should converge to a Standard method by definition. But this is not the case in figure 4. Is there any explanation, if this is the case?
Response 3: When t>Ek, \alpha is set to \epsilon. That is to say, the weight of peer networks’ correcting is \epsilon, which reaches its maximum. At Standard, it didn’t adopt any interchanging. So Co-correcting will not converge to Standard.
Point 4: In equation 13, it is not clear that "t" refers to the current epoch as defined later in algorithm 1.
Response 4: I add the explanation in algorithm 1 of “t” in lines 473
Point 5: Since we have smaller patches (images), did the authors try more larger batches with respect to R(t) ?
Response 5: I did some experiments and the batches have nothing to do with R(t). So I didn’t put it in my manuscript.
Point 6: In the prediction phase, which model is used, A, B or both? Is it possible to evaluate separately the performances of A and B?
Response 6: A and B just the names of two networks of Co-correcting, to convenient expression. We evaluate both A and B, and choose the best network as the final output.

Reviewer 2 Report (Previous Reviewer 3)
Regarding the additional answers provided by the authors, I have the following comments
1. Star catalogs are very accurate and I am not aware of stars missing from it. You should provide references to support your statement.
2. Star catalogs contain also brightness information about stars (I could assume this is what you indicate by "star shape" in the manuscript). Since all the specifications of the telescopes are available (including its sensitivity, i.e., the maximum magnitude/minimum brightness they can "see" on the image plane), the number and location of stars which will be image can be considered as a known information. To further stress this point, I want to highlight that star detection from images (for the sake of target segmentation) is a standard procedure also for cameras onboard satellites which has even more challenges than doing it with ground telescopes due to the orbital and rotational state uncertainty. See for instance, the paper below which explains how this has been done on board in the frame of the PRISMA mission for the sake of far range relative navigation with respect to a target spacecraft.
Benninghoff, H., Tzschichholz, T., Boge, T., & Gaias, G. (2013, May). A far range image processing method for autonomous tracking of an uncooperative target. In Proceedings of the 12th symposium on advanced space technologies in robotics and automation. Noordwijk, The Netherlands.
https://elib.dlr.de/86993/
3. You can see that star extraction is a standard procedure also in the manuscript below
Laas-Bourez, M., Coward, D., Klotz, A., & Boer, M. (2011). A robotic telescope network for space debris identification and tracking. Advances in Space Research, 47(3), 402-410.
Unfortunately, I cannot consider the authors' answer 3.2 to be adequate to the highlighted issue in my previous review.
The authors should better highlight and explain all these points in the manuscript before it can be accepted
Author Response
Response to Reviewer 2 Comments
Thanks for your valuable comments.
In your comments, you say that star detection is standard procedure. And I approval it. But our manuscript focuses on proposed a one-stage methods to detect objects from observation. We don’t deny the importance of star detection, but it is not the main focus of our work.
Our work tries to solve the noisy labels caused by the similarity of stars and objects, trying to introduce the semi-supervised learning to reduce the effect of noisy labels.
Your comments may right, but we concentrate on the different aspect.
Thanks again for your reviewing on this manuscript, I learn a lot from it.

This manuscript is a resubmission of an earlier submission. The following is a list of the peer review reports and author responses from that submission.
Round 1
Reviewer 1 Report
The training data of space debris contain a high number of noisy labels, which make to detect space debris become difficult. This paper investigate the lable-noise learning to space debris detection and present a novel label-noise learning paradigm.
In the Abstract, the authors write "Empirical experiments show that Co-correcting outperforms other state-of-the-art methods of label-noise learning in space debris detection. ", here you should describe which methods you compaired, specifying the name of the method.
In addition,
These methods are mainly used for cataloging and identification of newly added fragments, and these methods are not used for fragments that have already been catalogued. Therefore, the inspection of newly added fragments is very important. In the cataloging and identification of newly added debris, the most challenging is the cataloging and identification of the large amount of debris generated by anti-satellites.
See the following two papers:
"Investigating the risks of debris-generating ASAT tests in the presence of megaconstellations"
"Debris cloud of India anti-satellite test to Microsat-R satellite"
The paper should consider the large amount of debris generated by anti-satellites, how to use the paper's method for cataloging and identification, and how to prevent debris from being confused with stars and other debris.
Author Response
Response to Reviewer 1 Comments
see attached

Reviewer 2 Report
Dear Authors,
please, see the attachment.

Author Response
Response to Reviewer 2 Comments
see attached

Reviewer 3 Report
Introduction
Lines 32-33. Most of debris fragments are actually much smaller than 20 cm in LEO. One critical problem is that fragments smaller than a few cm cannot be detected by radars.
Related work. The selection of references must be more careful. References from 4 to 7 are not relevant to the debris detection problem from optical images, unlike wrongly stated by the authors. Please better clarify the content and the application scenario of the cited references.
Materials and methods
Please specify which are the noisy labelled images in figure 1.
Lines 202-203. This statement is not correct, GEO satellites do not produce such large amount of space debris. Most of them are intact. Space debris fragments are mostly produced by collisions and explosions in orbit.
Lines 237-242. This discussion should be revised to improve clarity, especially regarding the adopted symbology.
Lines 243-250. This paragraph is one example of many instances in the manuscript in which the same or similar information are repeated producing an unnecessary redundancy.
Figure 2. Please clarify whether the original image is synthetic or real.
How is the background defined to compute the threshold for binarization (equation 2).
Results
Lines 442-443. This point is not clear. I would expect the test data to be characterized by noisy labels.
The authors do not specify how training and testing images are generated.
General comment. Since debris are much closer than stars to the optical telescopes one critical way to separate them from stars is to exploit the fact that why will appear as stripes on the image plane due to their not negligible motion during the integration time of the image frame acquisition. The authors should carefully discuss this point to demonstrate the usefulness of the proposed approach
General comment. Another major aspect that the authors are not considering is that by knowing the pointing of the telescope in the inertial space, and thanks to the availability of star catalogues, the positions of stars in the images should be detected thus allowing to separate them from other objects. How do you comment on this point.
General comment. The quality of the English language should be significantly improved. Also, please avoid using not scientific terminology such as “precious/preciously” when referring to detection performance.
Author Response
Response to Reviewer 3 Comments
see attached

Reviewer 4 Report
In this paper, the authors presented a deep learning paradigm called "co-correcting" that was inspired from "co-teaching" presented in reference [18]. it was evaluated in the context of debris detection from remote sensing images.
General comments:
The manuscript presents some English writing errors:
in line 75:
"Hough transform".
In line 92:
"pre-proceedings" I guess it should be "pre-processing" ?
in line 230:
"methods base on" should be "methods based on" ...
line 238:
punctuation is missing in the end of the line "... by f_\theta(*) The .."
line 241:
"Label-noise learn problems" should be "learning problems" ?
line 240:
Check the notation "f_\theta"
in line 289 :
"transform consist of" should be "transform consists of"
Many other errors need to be verified by the authors. Also, please revisit the usage of the word "And" at the beginning of many sentences.
Technical comments:
The debris detection problem as defined is a class ambiguity problem and not a noisy data problem. I guess both co-teaching and JoCoR were applied to noisy data where we have mislabeled data and not a class ambiguity problem.
In mathematical morphology section the term "structure element" is reported which is more known as the "structuring element" ?
The co-correcting as proposed is simply based on interchanging p1 and p2 in the loss functions. So why it is supposed to be a co-correcting and not a co-erroring ? there is no guarantee that the predicted p1 is correct nor p2. What is the rationale behind this?
The problem will occur in the beginning of the training, when \alpha is small and p_i is not the expected class.
With t> Ek , the co-correcting should have no effect on the training as per equation 12, and should converge to a Standard method by definition. But this is not the case in figure 4. Is there any explanation, if this is the case?
In equation 13, it is not clear that "t" refers to the current epoch as defined later in algorithm 1.
Since we have smaller patches (images), did the authors try more larger batches with respect to R(t) ?
In the prediction phase, which model is used, A, B or both? Is it possible to evaluate separately the performances of A and B?
Author Response
Response to Reviewer 4 Comments
see attached

Round 2
Reviewer 1 Report
The paper can be accept in its present form.
Reviewer 2 Report
Dear Authors,
Thank you very much for addressing all my comments. The paper is well revised and I think that it can be accepted for publication.I want to tell you that I much appreciate your work, congratulations.
Reviewer 3 Report
Thanks for addressing my comments.
The manuscript clarity is improved.
Unfortunately, the answers to points 10 and 11 are still not adequate to justify the usefulness of the proposed methods.
As I already mentioned, the detection of stars and their removal from images collected by a ground optical telescope is a standard procedure and the need to know the specifics as dell’ as the position and attutire of the instrument cannot be seen as a critical aspect in this reviewer’s opinion.
consequently, although i agree that debris detection is an interesting problem I am still convinced that the motivation for the proposed approach is weak.
Author Response
Response to Reviewer 3 Comments
Thanks for your valuable comments.
Star removal is the standard procedure in traditional methods. And star catalog is widely used in star removal. However, in engineering context, star catalog can only provide the location of stars. In the original observation, we cannot completely remove the stars by their location without intensity of stars, leading to uncomplete removal.
Another problem is that the number of starts in catalog doesn’t math the original observation. For example, some stars in original observation are not catalogued in star catalog. And some stars may be missed by telescope, leading to wrong removing. Both situations will lead to unsatisfied star removal.
The main challenge is that the procedure of matching star catalog and observation is complex.
Based on the three challenges, we introduce label-noise learning to reduce the effect of stars.
